# Comparative Efficacy of Animal Depression Models and Antidepressant Treatment: A Systematic Review and Meta-Analysis

**DOI:** 10.3390/pharmaceutics16091144

**Published:** 2024-08-29

**Authors:** Piotr Ratajczak, Jakub Martyński, Jan Kazimierz Zięba, Katarzyna Świło, Dorota Kopciuch, Anna Paczkowska, Tomasz Zaprutko, Krzysztof Kus

**Affiliations:** 1Department of Pharmacoeconomics and Social Pharmacy, Poznan University of Medical Sciences, Rokietnicka 3, 60-806 Poznan, Poland; 2Institute of Human Genetics, Polish Academy of Sciences, Strzeszyńska 32, 60-479 Poznan, Poland

**Keywords:** selective serotonin reuptake inhibitors, fluoxetine, animal depression models, chronic unpredictable mild stress, rat strains, data availability, systematic review, meta-analysis

## Abstract

Background: Animal models are critical tools in the study of psychiatric disorders; however, none of the current models fully reflect human stress-related disorders, even though most of the knowledge about the mechanisms of depression comes from animal studies. Animal studies are useful in pharmacological research, whereby we can obtain results that translate into patient treatment by controlling environmental factors, especially in behavioural research. The authors systematically reviewed this issue since medical databases provide access to many primary studies. Methods: A systematic review and meta-analysis were conducted based on 25 primary studies. The studies were identified in databases such as PubMed, Embase, and Web of Science (December 2022) according to the inclusion and exclusion criteria established at the beginning of the research and published in the form of a protocol, following the PRISMA and Cochrane Collaboration methodology for secondary studies and CAMARADES (CAMARADES Berlin, QUEST-BIH Charité) for secondary studies on animals. Forest plot analyses were performed (data presented as Mean Difference, Random Model, Inverse Variance), Risk of Bias assessment (Systematic Review Center for Laboratory animal Experimentation (SYRCLE) evaluation), quality assessment of included studies (Animal research: Reporting of In Vivo Experiments (ARRIVE)), and a range of data from source publications were compiled in tabular form. The study analysed the popularity of both animal depression models (ADM) and rat strains used in pharmacological research to test the efficacy of antidepressant drugs based on the immobility time (IT) factor (Forced Swimming Test). The study examined selective serotonin reuptake inhibitors, namely fluoxetine, sertraline, paroxetine, citalopram, and escitalopram. Additionally, the study addressed issues concerning the “data availability statement” because precise IT data analysis was impossible in the case of 212 papers. Results: Our data confirm that the Chronic Unpredictable Mild Stress (CUMS) model is the most popular and versatile model used in preclinical depression research, while the two most popular rat strains were Wistar and Sprague-Dawley. The quality of included papers based on the ARRIVE assessment showed a ratio value equal to 0.63, meaning that studies were of intermediate overall quality. The Risk of Bias assessment based on the SYRCLE tool revealed a high risk related to the blinding and the random outcome assessment. In the meta-analysis, the results indicate that all analysed drugs demonstrated efficacy in reducing IT, and the most analysed drug was fluoxetine (confirmed based on 17 studies (19 models)). The analysis of the efficacy of ADMs showed that the most effective models were CUMS, Flinders Sensitive Line (genetic model), Social Isolation, Restraint Stress, and Low-dose Lipopolysaccharide (pharmacological model). Only 2.35% (5 out of 212) of corresponding authors responded to our data request. Conclusions: The study highlights the dominance of the CUMS model and the Wistar and Sprague-Dawley rat strains in preclinical depression research, affirming the efficacy of SSRIs, particularly fluoxetine, in reducing IT. The findings underscore the need for better data availability and methodological improvements despite intermediate overall study quality and notable bias risks. Enhanced transparency and rigorous assessment standards are essential for advancing the reliability of animal models in depression research.

## 1. Introduction

Depression is a mood disorder that affects the way a person thinks, feels, and functions. Individuals suffering from depression may experience deep sadness, hopelessness, loss of interest in daily activities, concentration problems, sleep, and appetite disturbances [1]. Depression may be chronic or recurring, significantly hindering a person’s capacity to perform at work or school and manage daily activities. The causes of depression are complex and multifactorial. They may include genetic predispositions, biochemical imbalances in the brain, traumatic life experiences, chronic stress, as well as other environmental and psychological factors. Some individuals may be more susceptible to developing depression due to hereditary genetic factors [2].

Depression is one of the most common mental disorders worldwide. The lifetime risk of experiencing depression is approximately 15–18%. Women are more susceptible to depression than men, which may be due to various factors, including hormonal, social, and psychological influences. Depression often co-occurs with other mental disorders, such as anxiety, and somatic illnesses, such as diabetes or heart disease [3].

Treatment for depression includes various approaches, depending on the severity and individual needs of the patient. The main treatment methods are antidepressant medications, such as selective serotonin reuptake inhibitors (SSRIs) or serotonin-norepinephrine reuptake inhibitors (SNRIs) and therapies, such as cognitive-behavioural therapy (CBT) and interpersonal therapy (IPT). Regular physical activity, a healthy diet, adequate sleep, and relaxation techniques such as meditation or yoga can support the treatment of depression [4].

Animal models of depression are a group of models utilised in major depressive disorder (MDD) studies, whether it is research about depression pathophysiology or the efficiency of antidepressant drugs. Animal models of depression can be arranged into two main categories: physically or genetically determined. Yet, in research utilising animal depressive models (ADM) to ensure a translation of obtained results to human patients, animal models should meet three requirements: high face validity, construct validity and predictive validity. These criteria ensure that the chosen animal models phenotypically resemble the disease in human patients and can alleviate disease symptoms with available medicaments [5,6]. Unfortunately, commonly used animal models are often flawed since they fail to meet the abovementioned criteria [6].

Chronic unpredictable mild stress (CUMS) is one of the most often utilised rodent models of depression. At its core, CUMS is one of the best models for representing the human depressive state [7], which works by exposing rats to mild stressors in an unpredictable manner for a few weeks [5,7]. These mild stressors include, among others, exposure to white noise, deprivation of food and water, cage tilting, social isolation, physical restraint, and many more [5]. Prolonged exposure to mild stressors eventually brings animals to a state in which they are unable to experience “pleasure”. However, this anhedonic state can be alleviated by chronic treatment with antidepressants [7]. These phenomena, in turn, confirm CUMS, as an appropriate model for the study of depression in humans. Nonetheless, the main problem with this model lies in the low repeatability of the results by different research teams, which can be attributed to many variables like the age of animals, painful stressors, different stress susceptibility based on animal strain, inadequate handling, and many more [7].

Conversally, the Flinders Sensitive Line (FSL) is a handbook example of a genetically determined animal model of depression. This strain of rats developed due to the selective breeding of Sprague-Dawley rats to increase their sensitivity to cholinesterase inhibitors [6]. FSL rats have a prominent reduction in motoric activity in both open field test and forced swim test, as well as enhanced REM sleep, reduced appetite, and lower overall immune system activity, making this model closely resemble human depression symptoms. Further data suggest that those symptoms can be alleviated by chronic treatment with antidepressant drugs as well as enhanced upon additional exposure to stressors [6,8]. Nonetheless, the FSL rats are flawed with changes in neurotransmitters, non-existent in human depression, rendering this model unfit for some research.

During the analysis of selected studies, apart from the models above, we also encountered the following animal models of depression: immobilisation, maternal deprivation, myocardial infarction, prenatal lipopolysaccharide (LPS) exposure, prenatal stress (PS), social defeat, social isolation (SI), restraint stress (RS) and thermoregulation. Those models are but a few examples of a highly diverse group of animal models of depression. Owing to the high diversity, animal models of depression can be further classified via the depression induction method, such as environmental induction, pharmacological induction, lesions, or genetic alterations via selective breeding. Each induction method, in turn, gave rise to the establishment of numerous animal models of depression. For instance, pharmacologically induced models include reserpine, corticosterone, and lipopolysaccharide-induced models, while environmentally induced models include: CUMS, chronic mild stress, maternal separation, chronic restraint stress and many more [6,9].

Serotonin (5-hydroxytryptamine, 5-HT) is a neurotransmitter that plays a crucial role in regulating mood, emotions, and sleep, and its deficiency is often associated with depression. Selective serotonin reuptake inhibitors (SSRIs) play a pivotal role in the treatment of depression [10]. All SSRIs were designed to selectively increase 5-HT activity by inhibiting the neuronal serotonin reuptake transporter (SERT). Despite similar antidepressant efficacy and indications, each SSRI has individual properties, including pharmacodynamics and pharmacokinetics. These differences contribute to diverse clinical indications, side effects, and drug interactions [10].

Introduced to the market in the late 1980s, SSRIs revolutionised the pharmacotherapy of depression, replacing older classes of antidepressants such as tricyclic antidepressants and monoamine oxidase inhibitors (MAOIs). Among SSRIs, the most commonly prescribed drugs include paroxetine, sertraline, fluoxetine, citalopram, and escitalopram [11].

Fluoxetine, a popular SSRI, is distinguished by its long half-life, allowing for less frequent dosing and reduced risk of withdrawal symptoms. It is also metabolised into the active metabolite norfluoxetine, which prolongs its therapeutic effect [12]. Fluoxetine inhibits serotonin reuptake by blocking the SERT, thereby increasing serotoninergic transmission and stimulating postsynaptic receptors. Simultaneously, it activates presynaptic receptors that inhibit serotonin release, delaying the full therapeutic effect [13].

Paroxetine is one of the strongest SSRIs with a high affinity for the SERT. Consequently, it exhibits potent antidepressant and anxiolytic effects, effectively treating various forms of depression and anxiety disorders [14]. Paroxetine’s mechanism of action involves prolonging serotonin’s action in the synaptic cleft by inhibiting its reuptake, enhancing and prolonging stimulation of postsynaptic serotonin receptors, leading to mood improvement after several days of use. However, paroxetine use may be associated with a higher risk of adverse effects such as weight gain, sexual dysfunction, and withdrawal symptoms due to its influence on other neurotransmitter systems [15].

Sertraline is an SSRI, often preferred for its favourable safety profile and relatively low risk of drug interactions. It is particularly beneficial for patients with comorbidities such as cardiovascular diseases [16]. Sertraline’s mechanism of action involves blocking serotonin reuptake into the nerve cell, thereby increasing serotonin levels in the central nervous system. It leads to enhanced serotoninergic neurotransmission.

Citalopram (SSRI) is effective in treating depression due to its high selectivity for the serotonin transporter. It is well tolerated and minimally affects other neurotransmitter systems, reducing the risk of adverse effects [17]. It works by blocking serotonin reuptake, increasing its concentration in the synapse and improving serotoninergic neurotransmission. Unlike other SSRIs, it has little to no effect on dopamine, norepinephrine, and GABA transport.

Escitalopram, the S-enantiomer of citalopram, is considered one of the most effective SSRIs. It exhibits high selectivity for the serotonin transporter, contributing to its efficacy and good tolerance profile [18]. Escitalopram is the newest and most potent drug, predominantly in serum, as it undergoes minimal metabolism. It shows negligible binding to serotonin, dopamine, adrenergic, and histamine receptors. Its high selectivity and minimal interaction with other neurotransmitter systems contribute to its popularity in clinical practice [19].

The study’s objective was to systematically review the efficacy of various ADMs and chosen SSRI drugs on depression-like behaviour based on the rat’s immobility-time (IT) factor observed in the forced-swimming test. Additionally, the study focused on the quality and credibility of the included primary studies and other factors like the popularity of rat strains and data access issues. The principal added value of this study is, to the best of our knowledge, the first comprehensive compilation of the efficacy and popularity of ADMs used in rats, alongside an analysis of the effectiveness of the most popular SSRI drugs (according to the number of records in the PubMed database).

## 2. Materials and Methods

### 2.1. Study Design

It is a systematic review with meta-analysis of animal studies based on the CAMARADES [20] and Systematic Review Centre for Laboratory Animal Experimentation (SYRCLE) [21] guidance as well as the PRISMA statement [22] and the Cochrane Collaboration methodology handbook [23]. The study protocol was registered at Open Science Framework and is available at https://osf.io/xvpq2 (Date created 22 December 2022; Date modified 2 January 2023).

### 2.2. Inclusion/Exclusion Criteria

Studies were included if they met the following criteria based on PICO:P: Animal studies: Studies involving rats (Rattus norvegicus) and investigating depression-like behaviour (various animal models of depression). Both adult and juvenile male rats were valid.I: Interventions: Only studies investigating the effects of sertraline, citalopram, escitalopram, fluoxetine, and paroxetine on depression-like behaviour of rats were included. The dose of the interventions has not been taken into account.C: Control group: Studies with a control group received either a vehicle, CMC, or NaCl and naïve were included.O: Outcomes: Only studies using the standard 5 min (300 s) observation of the forced swim test (FST) with the IT factor to measure depression-like behaviour were included.Language: Only English-written papers were included.

If studies did not meet the above criteria, they were excluded from the systematic review.

### 2.3. Search/Screening/Data Extraction

The literature search was conducted by a four-member research team consisting of PR, JM, JKZ, and KS and began on 8 December 2022. The databases used for the search were EMBASE, PubMed, and Web of Science (WOS). The search strategy was based on a combination of keywords: “depression”, “sertraline”, “fluoxetine”, “escitalopram”, “citalopram”, “paroxetine”, “rat”, and “forced swim test”, with the use of the logical operators AND and OR (more details in Appendix A). The search results were imported into the RYYAN website (rayyan.ai) to remove duplicates and manage the results.

The selection of studies took place in two stages: title and abstract review and full-text article review. Two independent reviewers assessed each article, and in case of discrepancies, decisions were resolved by consensus. The included studies evaluated the efficacy of sertraline, fluoxetine, escitalopram, citalopram, or paroxetine in the FST in rats, were published in English, and provided full access to the FST results data.

Data extraction was performed by all team members, with each article being assessed and verified by two reviewers to ensure data accuracy and reliability. Extracted data included bibliographic information (article title, authors, year of publication, publisher, journal, country, and corresponding author’s email address), funding and conflict of interest information (funding information and conflict of interest declarations), and study information (labels, tested substance, control substance, FST tool, FST results, rat strain, control group size, and statistical test used in the study).

The entire literature search, study selection, data extraction, and quality assessment process was meticulously documented, ensuring transparency and the possibility of replication by other researchers.

### 2.4. Data Availability Statement vs. Reality

In some publications, the results were presented only in a graph, and it was impossible to read their exact values. Using the official email address, between 3 January 2024, and 13 January 2024, 212 inquiries were sent requesting the numerical data for the IT values in the FST. The inquiry explained how the obtained information would be used and why it was impossible to get it from the publication.

### 2.5. SYRCLE

In the Risk of Bias (RoB) assessment, we utilised the SYRCLE tool [21]. Two independent reviewers reviewed each study, and any conflicts in assessments were resolved between them. Ten categories of risk of bias were evaluated using the “Low”, “Unclear”, and “High” scales. A “Low” rating indicates a low risk of bias, suggesting that the study is well-designed and conducted with minimal risk of systematic errors. An “Unclear” rating signifies an unclear risk of bias, indicating insufficient information for a definitive assessment of a particular aspect of the study. A “High” rating denotes a high risk of bias, suggesting significant methodological deficiencies in the study that may impact the reliability of its results.

### 2.6. Certainty of the Evidence—ARRIVE Instead of GRADE

#### 2.6.1. GRADE

The analysis of the certainty of evidence using the GRADE (Grading of Recommendations Assessment, Development, and Evaluation) was not conducted for several reasons:When conducting a GRADE assessment, it must be assumed that all the studies we analysed were non-randomised (classified as other studies), and therefore, these studies start with a score of 2 out of 4, which can be further upgraded or downgraded.Based on the SYRCLE analysis, it was determined that most of the analysed studies had a high Risk of Bias in terms of Blinding (Performance Bias), Blinding (Detection Bias), and Random outcome assessment (Detection Bias), which indeed translates to a significant downgrade based on those factors through the GRADE assessment.Conducted meta-analyses present high inconsistency due to the high heterogeneity of included studies (Higgins I^2^), another factor for downgrading these results.The overall meta-analysis results were consistent; however, subgroup data for citalopram and a few models (Immobilisation, Maternal Deprivation, Myocardial Infarction, Prenatal Stress, Social Defeat, Thermoregulation) were inconsistent and should be downgraded.

Based on the above, it can be concluded that the certainty of evidence (based on 25 included studies) is LOW or VERY LOW. Because of this, the Animals in Research: Reporting In Vivo Experiments (ARRIVE) tool should be used. While the ARRIVE guidelines are not explicitly focused on assessing “certainty” like GRADE, they play a crucial role in enhancing the quality of reporting in animal studies. This improved reporting facilitates a better assessment of the study’s validity and reliability, thereby indirectly contributing to the overall certainty of the research findings. Therefore, it is accurate that the ARRIVE guidelines help improve the transparency and thoroughness of animal research reporting, supporting a more confident interpretation of the results.

#### 2.6.2. ARRIVE

To examine the quality of analysed studies, we used the Animals in Research: Reporting In Vivo Experiments (ARRIVE) guidelines [24]. The ARRIVE guideline consists of 20 queries that check whether the study contains sufficient information about animal species, study design, randomisation, statistical analysis, the well-being of animals, etc. Two assessors answered each query on a 3-point scale: “2”—the provided data were clearly sufficient, “1”—the provided data were possibly sufficient and “0”—the provided data were clearly insufficient. Any discrepancies between the assessors’ ratings were discussed until a consensus was reached. If consensus was not achieved between the assessors, the third assessor was asked to resolve the conflict.

When the assessment of each query was concluded, a total quality ratio was calculated. It was done by taking all the scores from one query and comparing them to the maximum score that could have been achieved in the same category. The higher the ratio obtained, the higher the quality of a given category. That being said, the ratio value “>0.8” means that in the majority of studies, all of the required information in a given query has been found, classifying the category as “excellent”. Low ratio values “<0.5” mean that a given query was of “low quality” since most assessed studies lacked the required information. Ratio values in between indicated that studies had varying amounts of the necessary information, thus classifying the given query as “intermediate” quality.

### 2.7. Meta-Analysis Methodology

Data obtained during the systematic review were collected in an Excel spreadsheet (MS Office LTSC Standard 2021, version 2108) and subjected to meta-analysis using the Review Manager 5.1 software (Cochrane Collaboration). Two independent researchers verified the final results.

Since the data from the source publications were mean values of the IT, an analysis for continuous data considering values of Mean ± Standard Deviation was conducted. Results of the analysis were presented using the Mean Difference parameter (homogeneity of measurement scales, results were measured in the same units) and the confidence interval. Due to the high heterogeneity (Higgins I^2^) of the results included in the analysis, a random-effects model was applied. This model assumes that each study characterises a different population, leading to a different effect for each population (this model is indicated for high heterogeneity as it defines a larger confidence interval and allows for the determination of generalised effect measures). The inverse-variance method was used as the computational method. A 95% confidence interval was applied.

### 2.8. Sensitivity Analysis

Due to the high heterogeneity (Higgins I^2^) detected through both meta-analyses (SSRIs vs. Control (ADM) and Control (ADM) vs. Control), an assessment of sensitivity analysis was conducted using PQStat Software (2024) v.1.8.6.122. Poznan, Poland. The analyses showed that excluding every study (Appendix A) impacted the precision change factor however, in most cases, those values were small and similar. The most impactful were studies by Alam et al. [25] and Evans et al. [26] (Appendix A) and Dashti et al. [27] and Guo et al. [28] (Appendix A). The most probable reasons for the high heterogeneity in both analyses:Various types of ADMs were used in the included studies, leading to inherent heterogeneity that sensitivity analysis could not adequately address.Differences in study design include variations in drug dosages, duration of treatment, and experimental conditions.Differences in the methodologies used for the forced swimming test across the included studies.

### 2.9. Publication Bias

An assessment of publication bias was conducted using funnel plots generated with PQStat Software (2024), PQStat v.1.8.6.122. Poznan, Poland. Visual inspection of funnel plot studies distribution (symmetry/asymmetry) was performed and verified by the two independent researchers. The relationship between the effect size (Mean Difference MD) and standard error across the included studies was analysed. The points should be symmetrically distributed without publication bias around the mean effect size. For a more quantitative assessment, we conducted Egger’s statistical test to detect funnel plot asymmetry (Appendix A). This analysis is essential to assess whether the results of the studies might be influenced by the selective publication of positive findings, ensuring the reliability and validity of the overall conclusions.

## 3. Results

### 3.1. Literature Search Result

In the database search, we identified 2908 records. After eliminating duplicates, 1980 records were screened. From this, 426 articles were deemed relevant based on their titles and abstracts. Upon full-text screening, 25 articles met the inclusion and exclusion criteria and were included in the qualitative analysis. At the screening stage, studies were each verified by two independent reviewers, and the conflicts were resolved by reaching a consensus. The PRISMA flow diagram illustrating the search process for this systematic review is shown in Figure 1.

### 3.2. Data Availability

Out of 212 email inquiries (insufficient data (n = 207) + included studies (5 out of 25)—Figure 1) related to the FST results, we received six responses (2.35%), of which one paper had to be excluded from the analysis. Ultimately, five studies were included in the review (Figure 2).

### 3.3. Included Studies

In Appendix A, provided in the supplement, 25 studies that met the inclusion and exclusion criteria are presented and serve as the basis for the meta-analysis. While the table lists 25 studies, two [29,30] are represented twice due to using two different animal models of depression.

The table contains basic information such as the first author of the study, year of publication, country where the study was conducted, journal and publisher of the paper, any form of funding received, and conflict of interest. In subsequent columns, data on the animal model of depression used are presented, as well as basic information on study arms, rat strains and animal parameters, the FST setup investigated drugs, and the FST results.

According to the adopted criteria, studies that included the FST consisting of a pretest (the day before) and the primary test on the following day, with a measurement period of IT of 5 min (300 s), were included in the analysis. In some of the original results, the authors provided results in the form of mean ± SEM. Therefore, to standardise these results, the parameter standard error of the mean (SEM) was converted to standard deviation (SD), which was then used in the meta-analysis.

The results for five of the 25 included studies [29,31,32,33,34] were obtained through email contact with the authors (in these studies, the authors, for example, only provided figures with the IT results without specific numerical values).

As the primary aim of the systematic review was to compile results regarding IT, other studies conducted by the authors were not included in the table.

### 3.4. Models and Strains’ Popularity

We identified the CUMS model as the most popular, clearly indicating its significant prevalence (ten studies) [25,27,28,35,36,37,38,39,40,41] over other models (Figure 3). The next most popular model was RS, used in four experiments (in three papers) [29,30,42]. Other stress models, such as SI [26,43], Maternal Deprivation [34,44], and PS [45,46], were used twice. The rest of the models were used with equal and lower frequencies.

The analysis of the popularity of rat strains has shown that the Sprague-Dawley and Wistar strains are the two most frequently used in the field (Figure 4).

### 3.5. Quality Assessment

#### ARRIVE Table

Quality assessment results of 25 studies obtained under the ARRIVE guidelines are presented in Table 1. Of the 21 analysed queries, eight (1, 6, 7, 8, 9, 10, 12, 13) were classified as excellent, meaning that in most studies, all of the required information for those quarries was found. Three of the queries above (6, 8, 9) had perfect ratio scores, thus indicating that in those three categories, every research had essential data needed. Five inquiries (11, 14, 15, 18, 21) were classified as intermediate quality. The remaining eight queries (2, 3, 4, 5, 16, 17, 19, 20) had the lowest ratio values, meaning that a minority of studies met the requirements of these queries, thus classifying them as low quality. The carried-out assessment gave a total ratio value equal to 0.63, meaning that studies overall were of intermediate quality (Table 1).

### 3.6. Risk of Bias Assessment

To assess the risk of bias in individual studies, we utilised the SYRCLE’s Risk of Bias tool (Figure 5). Most studies (64%) had unclear risks regarding sequence generation. Conversely, 96% of the studies had well-balanced baseline characteristics, suggesting solid grounds for group comparisons. All 25 studies had an unclear risk of bias associated with allocation concealment. Additionally, a significant number of studies (72%) had unclear risks concerning random housing, indicating a need for a better description of these procedures. Most studies (76%) were at high risk of bias due to the lack of blinding (Performance bias). The high risk of bias in 96% of studies underscores significant issues with random outcome assessment. Approximately half of the studies (40%) had a low risk of bias in blinding (Detection bias). Most studies (64%) also had a low risk of bias related to incomplete outcome data. Nevertheless, all 25 studies had a low risk of reporting a selective outcome and other sources of biases, indicating the robustness of the research.

The assessment of the RoB using the SYRCLE tool revealed varied risks across different domains. Significant concerns regarding included in systematic review papers revolved around blinding (Performance and Detection Bias) and random outcome assessment, where most studies had a high risk of bias. On the other hand, baseline characteristics, selective outcome reporting, and other sources of bias were generally well-reported and had a low risk of bias.

### 3.7. Receptor Affinity of the Tested Drugs

Analysing the receptor affinity of various SSRIs, it becomes clear that each drug has a unique profile that may influence its therapeutic and side effects (Table 2). Fluoxetine (Prozac) shows an affinity for the receptors SERT, 5-HT2C, H2, α2, α3, and β4. This broad range of affinity can contribute to its antidepressant efficacy but also to various side effects resulting from interactions with multiple receptor types. Paroxetine (Seroxat) binds to the receptors SERT, D2, 5-HT1, 5-HT2, 5-HT3, H1, α1, α2, and β, reflecting its strong action on serotonin reuptake and has a significant impact on various serotonin receptor subtypes, which may explain its efficacy in treating anxiety disorders and depression. Sertraline (Zoloft) targets the receptors SERT and σ1, which may contribute to its therapeutic properties. Citalopram (Cipramil) shows a more focused affinity for the receptors SERT and H1, which may limit its side effect profile compared to other SSRIs. Escitalopram (Lexapro) exhibits affinity for the receptors SERT, D2, 5-HT1A, 5-HT2A, 5-HT2C, H1, M1, and α2, combining strong serotonin reuptake inhibition with additional modulatory effects on various serotonin and histamine receptors. These differences in receptor affinity not only help to understand the clinical efficacy of these drugs but also provide insights into their diverse side effect profiles, enabling the personalisation of treatment strategies for patients with depression and related disorders.

### 3.8. Meta-Analysis

#### SSRI Efficacy

The results from the meta-analysis indicate that all analysed drugs demonstrated efficacy in reducing the IT (Figure 6). The results are presented based on the division according to the applied drug.

The efficacy of fluoxetine was confirmed based on 17 studies (19 models) [25,26,27,28,29,30,33,36,38,39,40,41,42,43,45,46,47]. The analysis compared to the model control group (animals with induced depression model, without drug administration) showed a positive Mean Difference (analysed by Inverse Variance) at the level of IV-MD −46.52 [−59.76, −33.28] *p* < 0.00001 (95% CI), I^2^ = 95%, indicating the drug’s efficacy in the analysed range.

Efficacy in this range was also demonstrated for other analysed substances:citalopram (2 studies [32,44]; IV-MD −23.55 [−31.29, −15.80] *p* < 0.00001 (95% CI), I^2^ = 0%),escitalopram (3 studies [31,34,48], IV-MD −31.09 [−53.07, −9.12] *p* = 0.006 (95% CI), I^2^ = 56%),paroxetine (2 studies [35,49], IV-MD −66.30 [−87.46, −45.14] *p* < 0.00001 (95% CI), I^2^ = 0%),and sertraline (1 study [37], IV-MD −74.22 [−107.05, −41.39] *p* < 0.00001 (95% CI)).

Positive results for the analysed SSRIs were also obtained in the overall analysis—IV-MD −45.33 [−56.21, −34.45] *p* < 0.00001 (95% CI), I^2^ = 94% (Figure 6).

### 3.9. Animal Depression Model Comparison

The analysis of the efficacy of ADM showed that the most effective models are CUMS, FSL, Prenatal LPS Exposure, SI, and RS (Figure 7).

The efficacy of the CUMS model was confirmed based on ten studies [25,27,28,35,36,37,38,39,40,41] comparing the IT parameter values in the model control group with the non-model control group. It showed an IV-MD value of 75.93 [48.95, 102.91] *p* < 0.00001 (95% CI), I^2^ = 97% in favour of the non-model group, confirming the efficacy of the applied model in increasing the IT (the studied drugs subsequently reduce this parameter).

Similar results were obtained for the:FSL model (1 study [49], IV-MD 81.00 [42.95, 119.05] *p* < 0.0001 (95% CI)),Prenatal LPS Exposure (1 study [33], IV-MD 41.75 [15.40, 68.10] *p* = 0.002 (95% CI)),SI (2 studies [26,43], IV-MD 81.03 [76.44, 85.62] *p* < 0.00001 (95% CI), I^2^ = 0%),and RS (3 studies/4 models [29,30,42], IV-MD 35.29 [17.29, 53.28] *p* = 0.0001 (95% CI), I^2^ = 64%).

For the models of Immobilisation [29], Maternal Deprivation [34,44], Myocardial Infarction [31], Prenatal Stress [45,46], Social Defeat [47], and Thermoregulation [32], no statistically significant results were obtained to confirm the efficacy of the methods.

In contrast, the overall analysis showed that ADMs are effective in the IT increase—IV-MD 49.58 [34.60, 64.57] *p* < 0.00001 (95% CI), I^2^ = 96% (Figure 7).

### 3.10. RoB Assessment

An assessment of the funnel plot regarding the results of SSRI dugs efficacy showed that some of the studies were outside the confidence interval (CI), however, non-asymmetric distribution was detected (Figure 8). The results of Egger’s test indicated no significant evidence of publication bias (*p* = 0.818736).

Further assessment of funnel plots regarding the results of ADMs showed that multiple studies were outside the confidence interval (CI), however, non-asymmetric distribution was detected (Figure 9). The results of Egger’s test indicated no significant evidence of publication bias (*p* = 0.287667).

## 4. Discussion

Depression is a debilitating mental health disorder that affects millions of patients worldwide, so understanding the antidepressant treatment efficacy is crucial for developing the best possible solutions for this disease. One of the best possible methods used in this scientific area is conducting animal pharmacological studies. Those studies can be analysed based on artificial environments and controlled factors, creating a background for further clinical studies. One of the best examples of the methodology used in those studies is the FST—a commonly used behavioural test for evaluating the antidepressant properties of drugs, which allows for observing changes in animal behaviour that may reflect the effects of antidepressant medications.

Our study aims to demonstrate the efficacy of SSRIs and the validity of different models used in depression research. Additionally, this review provides insights into the quality and reliability of the analysed studies. We presented various data points that offer a comprehensive view of how such studies should be conducted and where they are typically published. This detailed analysis underscores the efficacy of the drugs and models as well as highlights the methodological rigour required for reliable research in this field.

### 4.1. The Efficacy of SSRIs

SSRIs have been proven to be effective in patients with depression and anxiety disorders. Cipriani et al. [50] showed in a meta-analysis of 21 antidepressant drugs (including all the drugs analysed in our study) that each of the tested substances showed to be effective in patients with depression. SSRIs bind mainly to the SERT transporter, preventing the reuptake of serotonin from the synapse back into the presynaptic neuron. Blocking the reuptake increases serotonin concentration in the synaptic space, which enhances signalling through postsynaptic receptors [51]. SSRIs are generally well tolerated but can cause some side effects. The most common include nausea, insomnia, dry mouth, dizziness, and decreased libido. SSRIs are considered safe for long-term use, although some studies suggest prolonged use may affect weight and bone density [52].

The meta-analysis showed that all tested drugs were effective in various ADMs described in our study. However, an essential factor that needs to be considered while assessing the results is that most of the included studies in our meta-analysis described the use, safety, and efficacy of fluoxetine. At the same time, other medications were analysed based on a much smaller number of papers. That is why fluoxetine results could be the most precise even though other drugs showed more significant Mean Difference results (such as sertraline or paroxetine).

Paroxetine is a medication used in the treatment of depression and other psychiatric disorders such as anxiety disorders, obsessive-compulsive disorder (OCD), post-traumatic stress disorder (PTSD), and panic disorder. We found that it is one of the most effective medications in the study; however, this assumption was based only on the few available studies. On the other hand, paroxetine’s effectiveness has been confirmed multiple times. Sawamura et al. [53] confirmed that paroxetine reduced hypervigilant behaviour during the task session in rats. At the same time, Qiu, Hong-Mei et al. [54] demonstrated that rats receiving paroxetine showed a significant reduction in depressive behaviours, a decrease in malondialdehyde levels, an increase in superoxide dismutase and catalase activity. In contrast, the expression of serotonin and norepinephrine transporters was increased. Additionally, Keller et al. [55] confirmed its clinical use in the treatment of major depression in adolescents. In a meta-analysis conducted by Barbui et al. [56], researchers confirmed that patients receiving paroxetine scored better than the placebo group.

We found that sertraline is the most effective medication among those included in the meta-analysis; however, like paroxetine, the small number of studies (only one meeting our criteria) may influence the result’s precision. Sertraline has well-documented therapeutic effects in treating depression in both rats and humans. Mukherjee et al. [57] confirmed the efficacy of sertraline in a rat model, where it reduced depressive activity in the tested rodents. A meta-analysis conducted by Kishi et al. [58] confirmed the efficacy of sertraline in patients with major depressive disorder. However, the use of sertraline was associated with a higher incidence of side effects, such as nausea and vomiting, and a higher rate of treatment discontinuation due to adverse effects.

Fluoxetine is one of the most commonly prescribed antidepressants in the world. It is used to treat various psychiatric disorders, and it is widely used in the treatment of major depressive disorder, reducing symptoms of depression such as sadness, loss of interest, fatigue, and sleep disturbances. Fluoxetine is also used in the treatment of OCD and anxiety disorders. Kryst et al. [59] examined the effects of chronic fluoxetine treatment on anxiety and depressive behaviours in rodents and confirmed its efficacy based on meta-analysis. However, the researchers noted that chronic administration of fluoxetine might increase anxiety in animals during adolescence. Bahji et al. [60], also based on meta-analysis, confirmed that fluoxetine treatment is effective compared to placebo in the treatment of bipolar depression.

Our meta-analysis showed that citalopram is an effective treatment in animal models of depression. However, as with paroxetine, its effect was confirmed based only on two studies. Nevertheless, Overstreet et al. [61] confirmed the antidepressant and anxiolytic effects of the medication in animal studies. Moreover, Cui et al. [62] confirmed in a meta-analysis that the clinical efficacy of citalopram is similar to other SSRIs, but citalopram acts faster. Additionally, adverse events during citalopram administration are less frequent and of lower intensity. The faster action of citalopram compared to other SSRIs was also confirmed in a study conducted by Hsu et al. [63], who demonstrated that compared to sertraline, citalopram tended to have better efficacy in treatment and fewer side effects.

Escitalopram is another popular drug for treating depression and anxiety disorders. Seo et al. [64] confirmed that administering escitalopram to rats reduces the IT in the FST in the chronic restraint stress model. Additionally, in a meta-analysis by Kennedy et al. [65], escitalopram showed significant superiority in efficacy. The overall treatment effect was better compared to all the comparator drugs analysed in the study, including citalopram, fluoxetine, paroxetine, and sertraline.

### 4.2. Animal Depression Models

The conducted meta-analysis has proven that five out of ten animal models of depression are suitable for depression research. The most effective model is CUMS. During this paradigm, rats are exposed unpredictably to mild stressors over time, which are supposed to resemble daily disturbances experienced by humans [9].

There is a plethora of stressors used in establishing CUMS model. Yet, due to the lack of uniform protocol, independent research teams often utilise different stressors in establishing CUMS for their studies. For example, studies by Alam et al. [25] and Zavvari et al. [39], both used CUMS paradigm to establish a depressive state in rats, yet to achieve this utilised different sets of mild stressors, such as white noise exposure, restraint stress, grouped housing, etc. Antoniuk et al. [66] published a meta-analysis surveying the CUMS model’s reliability. One of the analyses tries to evaluate the usage of individual stressors in various studies. The analysis revealed that food and water deprivation, present in approximately 12% of studies analysed, was the most commonly used stressor in establishing CUMS in rats. Other frequently used stressors, in descending order, included modified light/dark cycle, wet bedding, tilted cage, social stress, forced swimming, and restraint [66]. These discrepancies among stressors allude to low replicability of results across different research teams.

CUMS paradigm leads to a depressive-like state in rodents by inducing “anhedonia” [7,67], the presence of which is also utilised to diagnose MDD in humans [68]. This state in rodents is understood as a decrease in recognition of novel objects, a deterioration in self-grooming and sucrose consumption, and in animal interactions [5,6,9,67]. Our meta-analysis showed that in each study utilising CUMS paradigm, the IT values in the FST for rats with induced depression were at least twice as high as in control groups. Rats under CUMS paradigm also show similarities with the MDD individuals on molecular levels. For instance, Wang et al. [69] showed increased levels of corticotropin-releasing hormone (CRH) [69,70], Grippo et al. [71] showed elevated levels of IL-1β and TNF-α and López-López et al. [72] alterations in levels of reactive oxygen species (ROS) and total antioxidant capacity in modelled rats.

The following is the Restraint Stress model, in which rodents are placed in translucent tubes, restricting their movement for at least an hour. This procedure, repeated over weeks in various studies [6,9,69], aims to simulate daily stressful situations that can induce a depressive state [9]. Mao et al. [73], based on a meta-analysis, found that chronic restraint stress (CRS) was more successful in introducing “anhedonic” behaviour in rats rather than in other rodent species. Our meta-analysis indicates a potential exhibition of despair-like behaviour in rats subjected to CRS/RS. Our findings can be further confirmed by numerous research studies utilising this model in despair-like behaviour evaluation tests, yet these have not been included in our study [74,75,76]. The CRS/RS paradigm has also contributed significantly to discoveries in the MDD area. Chiba et al. [76] revealed that alterations in glucocorticoid receptor (GR) and brain-derived neurotrophic factor (BDNF) are observed in rodents exposed to restraint stress. CRS/RS rodents were one of the models used in numerous research regarding the hypothalamic-pituitary-adrenal (HPA) axis [77,78].

Flinders sensitive line (FSL) is a genetically determined animal model of depression [79], obtained via selective breeding of Sprague-Dawley rats, which showed increased sensitivity to di-isopropyl fluorophosphate, a cholinesterase inhibitor. Cholinergic super-sensitivity is also observed in individuals suffering from MDD [80]. This model closely replicates most of the core symptoms associated with MDD, such as psychomotor retardation, reduced appetite and body weight, decreased activity of the immune system, changes in the REM sleep phase and anxiety-like behaviour [81]. Elfving et al. [82] also reported that FSL rats exhibit lower levels of BDNF in the hippocampus yet higher levels in serum. Studies utilising the FSL paradigm also indicated alterations in GABA levels [81], akin to MDD patients [83]. Yet FSL rats are unfit for some aspects of depression studies since this strain is flawed with significant neurotransmitter alterations compared to the MDD patients [6,81]. During the meta-analysis, we included two studies utilising FSL (one without a control group), meaning we could not conduct a proper meta-analysis for this model. However, other authors [79,84] have also confirmed the results presented in our study.

Social Isolation (SI), capitalises on rodents’ natural inclination as herd animals need social interaction [6]. In the model, rats are isolated from others for an extended period [85]. This approach primarily aims to simulate social stress and loneliness, factors known to contribute to depression in humans. Our analysis dubbed the social isolation model as an effective model in MDD research, yet once again, our analysis was limited by a few included studies. Nonetheless, data from independent studies using the social isolation paradigm and the FST further confirm our findings [86,87,88]. In biochemical studies, Begni et al. [85], based on SI rats, showed significant downregulation of BDNF. Similar results, but with C57BL/6J mice, were achieved by Ieraci et al. [89].

Lipopolysaccharide (LPS)-induced depression is an animal model that mimics depression by triggering immune system activation. It leads to the release of pro-inflammatory cytokines that cross the blood-brain barrier, causing neuroinflammation and depression-like behaviour in rodents [90,91]. An extension of the paradigm of using LPS to induce depression is the prenatal administration of LPS to induce depression in the offspring by affecting brain development due to the neuroinflammatory processes [33]. LPS-induced depression in rodents is manifested by several behavioural changes, including a reduction in locomotor activity, alterations in sleep patterns, social withdrawal, and formation of anhedonic and despair-like behaviour [91]. Our meta-analysis included only one study using LPS to induce prenatal depression, which found the model effective but requires further confirmation. Custódio et al. [92] conducted a study in which they utilised the prenatal LPS exposure paradigm on Swiss mice and later evaluated depression-like behaviour using the FST, observing higher IT in rodents after the prenatal LPS exposure paradigm, comparable with our findings. A study by Lin et al. [93] showed that rats prenatally exposed to LPS significantly reduced 5-HT, which corresponds to the MDD cases. Furthermore, dopamine levels also decreased, which can be observed as well in depressed individuals [94]. Additionally, this research further indicates the good face validity of this model since, in the dorsal hippocampus of LPS-treated rats, lower levels of glucocorticoid receptors were observed, akin to the MDD patients [95].

### 4.3. Animal Strains

Choosing proper rat strain in depression research is crucial for obtaining consistent and reliable results. Each model has unique advantages and limitations, and their use should be carefully tailored to the specific research objectives.

Wistar rats are popular in depression research due to their consistent responses to various stressors and medications. Their physiology and biochemistry are well-studied, facilitating the interpretation of research results. Additionally, Wistar rats are characterised by a mild temperament, making handling and behavioural experiments easier. WKY rats, a subtype of Wistar, exhibit extremely passive behaviour in the FST and are resistant to antidepressant medications, making them particularly useful in studies of treatment-resistant depression mechanisms [96]. Sprague-Dawley rats are another important model in depression research. These rats are often used in behavioural studies due to their calm nature, facilitating handling and observation. Sprague-Dawley rats also have well-documented physiology and biochemistry, making them valuable models in various preclinical studies. Their reactions to different stressors and medications are well-studied, making them useful in investigating depression mechanisms and testing new antidepressant drugs. Moreover, Sprague-Dawley rats are characterised by rapid growth and high survival rates in laboratory conditions, making them ideal for long-term experiments [97].

Flinders Sensitive Line (FSL) and Flinders Resistant Line (FRL) were genetically selected based on their responses to muscarinic drugs. FSL rats exhibit depression-like behavioural traits, such as decreased activity and anhedonia, making them valuable models in depression research. FRL rats, on the other hand, are more resistant to stress and exhibit fewer depressive symptoms, allowing comparisons between strains and identification of mechanisms of depression resistance [96]. FSL and FRL models are used to study the genetic and neurochemical foundations of depression and to test the efficacy of new antidepressant drugs. Their genetic characteristics allow for precise studies of the molecular mechanisms associated with depression.

Lewis rats are frequently used in immunological and behavioural research. They are known for their greater susceptibility to autoimmune diseases, making them useful in studies of the pathogenesis of these diseases. In the context of depression research, Lewis rats exhibit unique responses to stressors, making them interesting models for studying the interactions between stress and the immune system. Lewis rats tend to show an increased response to stress, which can lead to the development of depressive symptoms. Their use in depression research helps to understand how stress affects the immune system and how these interactions may contribute to the development of depression [98].

The most common target of SSRI drugs is the serotonin transporter (SERT). Among the serotonin receptors affected by SSRIs, the most frequently mentioned were 5-HT2C, 5-HT1A, and 5-HT2A. The influence on other receptors, such as NET and DAT, was reported less often, indicating a more specific action of SSRIs in the context of depression.

### 4.4. Data Availability

Data availability was one of the main issues authors faced during the systematic review process. There were instances where FST data were not available. Despite efforts, such as contacting authors via email (based on the data availability statement) for data access, those requests were mainly unsuccessful, indicating potential limitations in data transparency and accessibility within the literature. 

The lack of response underscores the need for better data management practices and open communication within the research community to support study replication and fully utilise collected data. In the broader context of meta-analyses, data availability issues often hinder comprehensive synthesis and introduce potential biases [99].

### 4.5. Limitations

Several important limitations of our study must be considered. First, the search was conducted in only three databases, which may have omitted some relevant studies. Despite a thorough search of publications databases, there is a risk that some available studies or unpublished data were not identified, which may impact the completeness of our analysis. Additionally, we applied restrictive criteria regarding the Forced Swim Test (FST), which may have greatly limited the number of included studies. Another important factor that significantly oversimplified the view on the animal model of depression is the exclusion of other objective parameters commonly used in the field of animal studies to confirm the effectiveness of depression models. One of the most important examples of this would be anhedonia, which was not analysed in our systematic review. By narrowing the inclusion criteria to only one parameter (immobility time), the perspective on the models and their response to treatment was simplified. Other limitations were the specific parameters, such as the sex of the animals and the selection of drugs, where fluoxetine (FLU) was the most commonly used drug, potentially influencing the results. The substantial limitation was also related to the data availability issue described above. Most of the analysed studies (76%) were assessed as having a high risk of bias due to the lack of blinding, which could affect the reliability of the findings.

## 5. Conclusions

This study reaffirms the efficacy of selective serotonin reuptake inhibitors (SSRIs) in animal models of depression, particularly highlighting the Chronic Unpredictable Mild Stress (CUMS) model’s efficacy. Our findings underscore the intermediate quality of the included studies and reveal significant risks of bias, particularly concerning blinding and outcome assessment. The data also emphasise the necessity for improved transparency and data availability in preclinical research. By addressing these methodological concerns, future studies can enhance the reliability and applicability of animal models in understanding and treating depression. Enhanced data management and open communication within the research community are essential for advancing this field.

## Figures and Tables

**Figure 1 pharmaceutics-16-01144-f001:**
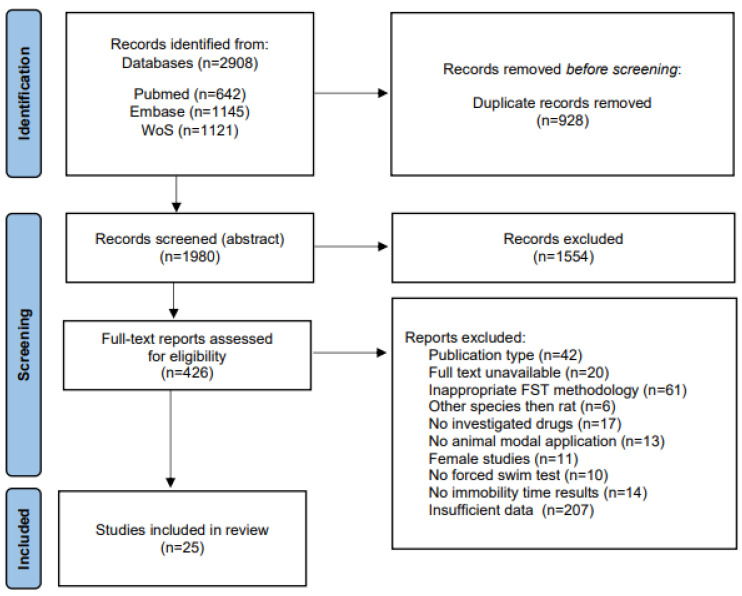
PRISMA flow diagram.

**Figure 2 pharmaceutics-16-01144-f002:**
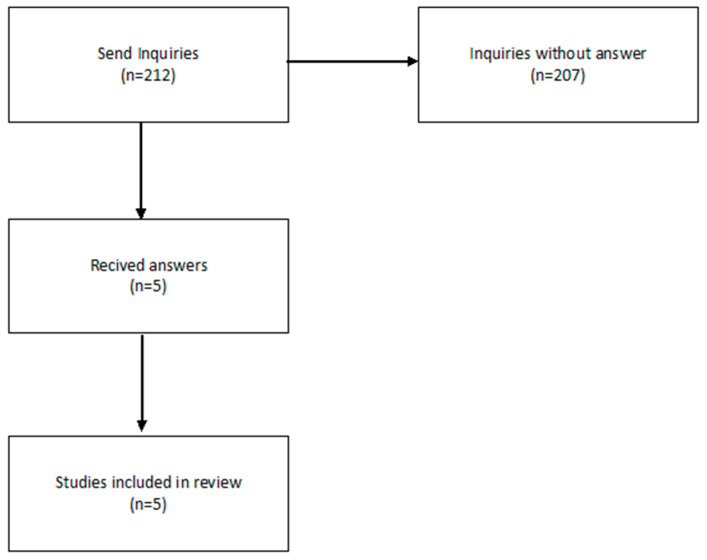
Realisation of data availability statement.

**Figure 3 pharmaceutics-16-01144-f003:**
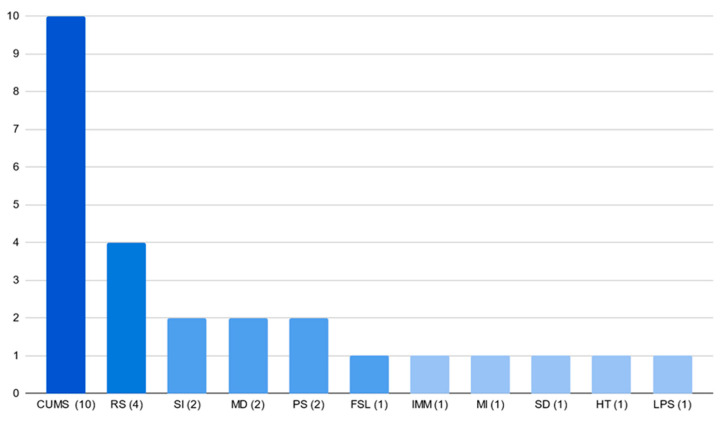
The most frequently used animal models of depression in analysed papers.

**Figure 4 pharmaceutics-16-01144-f004:**
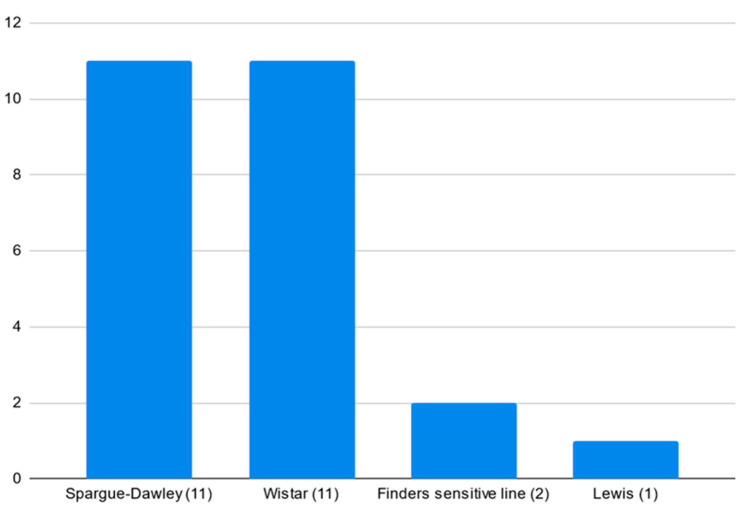
The most frequently used rat strains in analysed papers.

**Figure 5 pharmaceutics-16-01144-f005:**
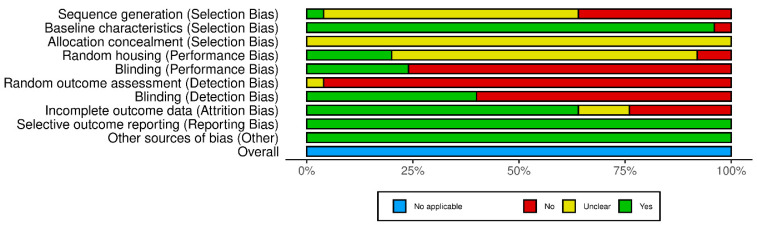
SYRCLE analysis.

**Figure 6 pharmaceutics-16-01144-f006:**
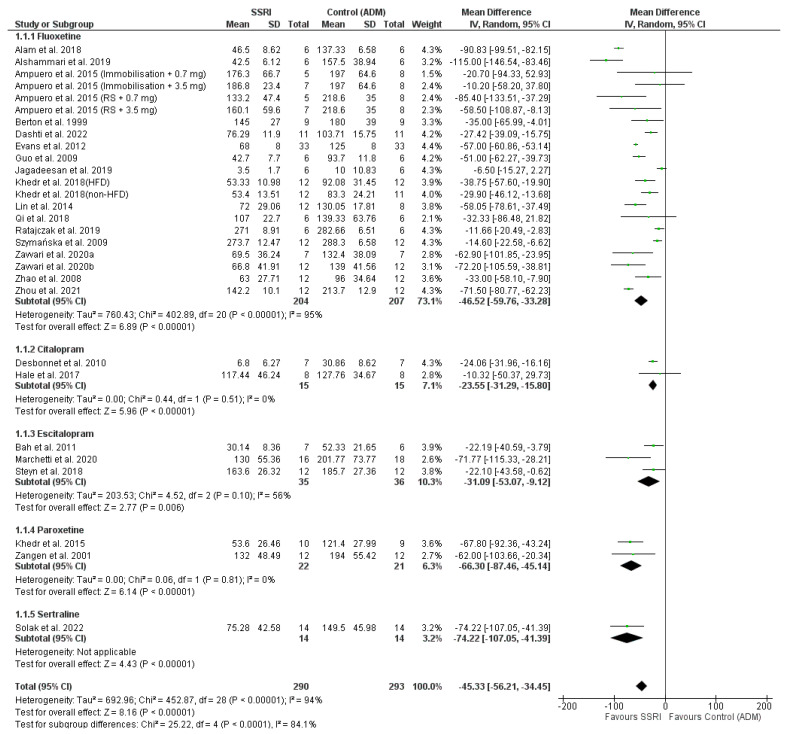
Forest-plot of SSRIs efficacy [25,26,27,28,29,30,31,32,33,34,35,36,37,38,39,40,41,42,43,44,45,46,47,48,49].

**Figure 7 pharmaceutics-16-01144-f007:**
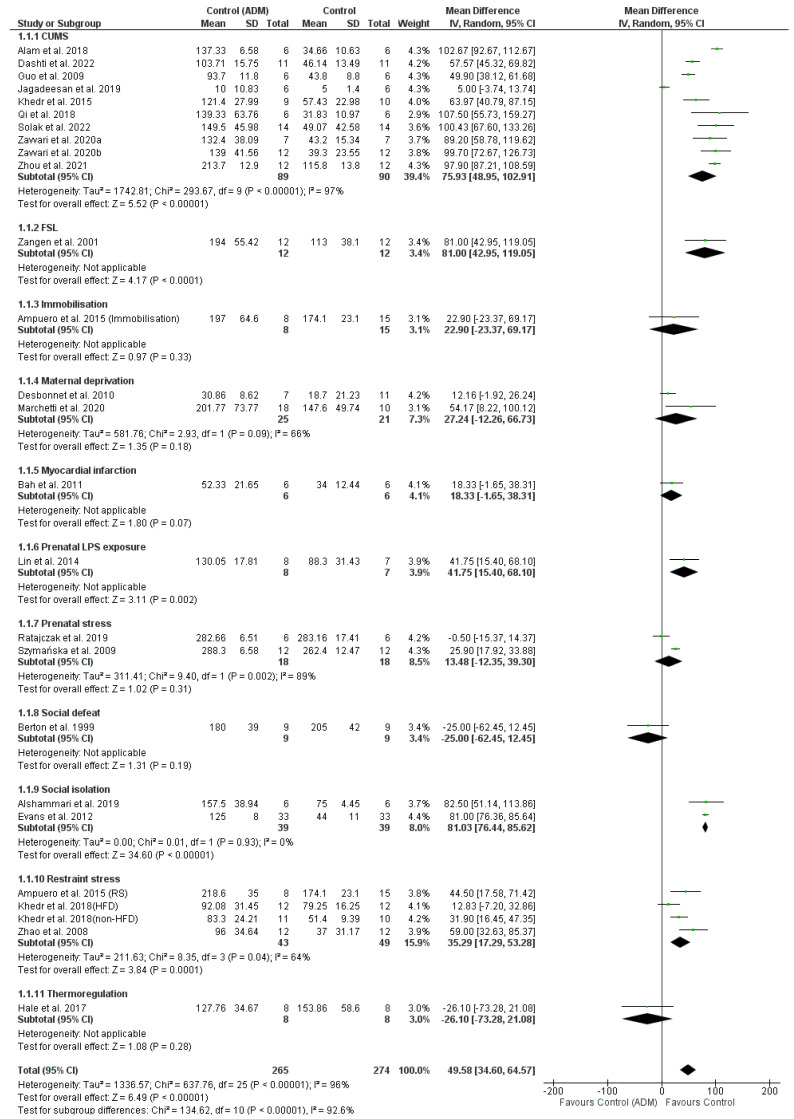
Forest-plot of ADMs efficacy [25,26,27,28,29,30,31,32,33,34,35,36,37,38,39,40,41,42,43,44,45,46,47,48,49].

**Figure 8 pharmaceutics-16-01144-f008:**
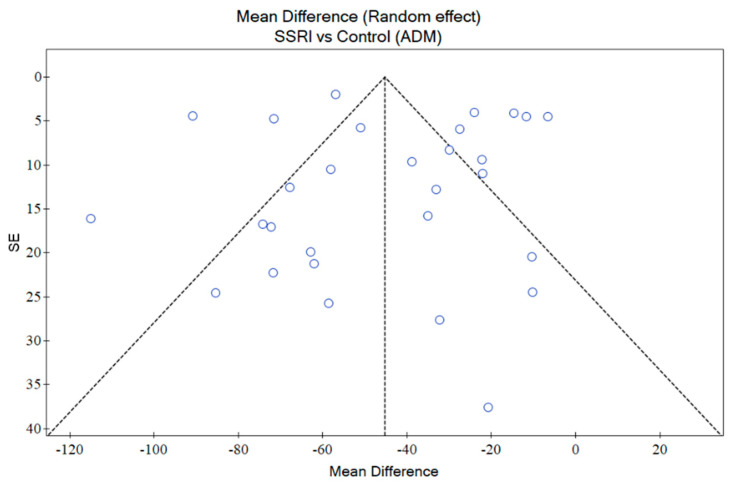
Assessment of publication bias based on SSRIs efficacy data.

**Figure 9 pharmaceutics-16-01144-f009:**
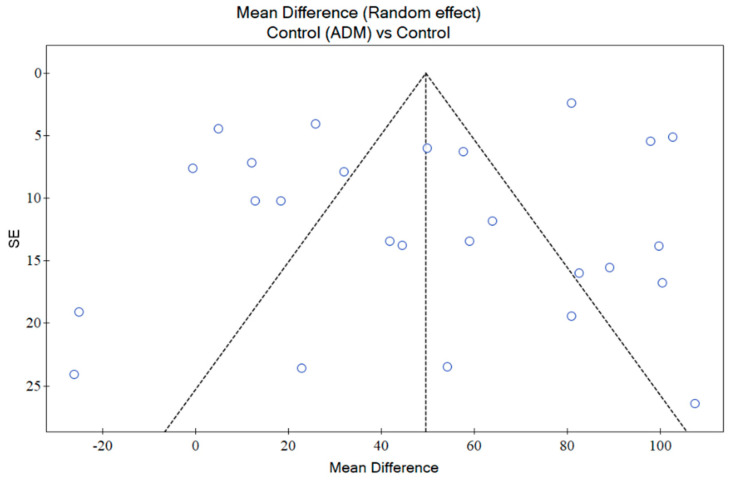
Assessment of publication bias based on ADMs efficacy data.

**Table 1 pharmaceutics-16-01144-t001:** ARRIVE analysis.

	Authors and Year	Arrive Item																					Total
		1	2	3	4	5	6	7	8	9	10	11	12	13	14	15	16	17	18	19	20	21	
	Alam et al. (2018) [25]	2	1	1	0	0	2	2	2	2	2	0	1	2	2	1	0	1	1	0	0	2	24
	Alshammari et al. (2019) [43]	2	1	1	0	0	2	2	2	2	2	1	2	2	2	2	0	1	1	0	2	2	29
	Ampuero et al. (2015) [29]	2	1	1	0	1	2	2	2	2	2	1	2	2	2	1	1	1	1	0	0	2	28
	Bah et al. (2011) [31]	2	1	1	0	2	2	2	2	2	0	1	2	2	2	1	0	1	1	0	0	1	25
	Berton et al. (1999) [47]	2	1	1	0	1	2	2	2	2	2	1	2	2	1	2	1	1	1	0	0	0	26
	Dashti et al. (2022) [27]	2	1	1	0	0	2	2	2	2	2	1	2	2	2	1	1	1	2	0	2	2	30
	Desbonet et al. (2010) [44]	2	1	1	0	1	2	2	2	2	2	1	2	2	2	2	1	1	1	0	2	0	29
	Evans et al. (2012) [26]	2	1	1	0	1	2	2	2	2	1	1	2	1	2	2	1	1	1	0	2	2	29
	Guo et al. (2009) [28]	2	1	1	0	2	2	2	2	2	2	1	2	2	1	2	0	1	1	0	0	0	26
	Hale et al. (2017) [32]	2	1	1	0	1	2	2	2	2	0	2	2	2	2	2	1	1	1	0	0	2	28
	Jagadeesan et al. (2019) [41]	2	1	1	0	0	2	2	2	2	2	2	2	2	2	2	1	1	1	0	2	2	31
	Khedr et al. (2015) [35]	2	1	1	0	0	2	0	2	2	2	2	2	2	1	2	0	1	1	0	0	2	25
	Khedr et al. (2018) [30]	2	1	1	0	0	2	2	2	2	2	1	1	1	1	0	0	1	1	0	2	2	24
	Lin et al. (2014) [33]	2	1	1	0	0	2	2	2	2	0	1	2	2	2	1	0	1	1	0	1	2	25
	Marchetti et al. (2020) [34]	2	1	1	0	1	2	1	2	2	0	1	2	2	1	1	0	1	1	0	2	2	25
	Qi et al. (2018) [36]	2	1	1	0	0	2	2	2	2	2	1	2	2	1	2	0	1	1	0	0	2	26
	Ratajczak et al. (2019) [46]	2	1	1	0	0	2	2	2	2	2	1	2	2	1	2	1	1	1	0	0	0	25
	Solak et al. (2022) [37]	2	1	1	0	0	2	2	2	2	2	2	2	2	2	2	1	1	1	0	2	2	31
	Steyn et al. (2018) [48]	1	1	1	0	1	2	2	2	2	2	2	1	2	2	1	1	1	1	0	0	2	27
	Szymańska et al. (2009) [45]	2	1	1	1	0	2	1	2	2	2	1	2	2	1	2	0	1	1	0	0	2	26
	Zavvari et al. (2020) [39]	2	0	1	0	0	2	2	2	2	2	1	2	2	2	1	1	1	1	0	0	2	26
	Zaavari et al. (2020) [38]	2	1	1	0	0	2	2	2	2	2	1	2	2	2	1	0	1	1	0	0	2	26
	Zangen et al. (2001) [49]	1	1	1	0	0	2	1	2	2	2	1	2	1	1	1	0	1	1	0	0	0	20
	Zhao et al. (2008) [42]	2	0	1	1	1	2	2	2	2	2	1	2	2	1	2	0	1	1	0	0	0	25
	Zhou et al. (2021) [40]	2	1	1	0	1	2	2	2	2	2	1	2	2	1	1	1	1	1	0	2	2	29
Category score		48	23	25	2	13	50	45	50	50	41	29	47	47	39	37	12	25	26	0	19	37	665
Maximum score		50	50	50	50	50	50	50	50	50	50	50	50	50	50	50	50	50	50	50	50	50	1050
**Ratio**		**0.96**	**0.46**	**0.50**	**0.04**	**0.26**	**1.00**	**0.90**	**1.00**	**1.00**	**0.82**	**0.58**	**0.94**	**0.94**	**0.78**	**0.74**	**0.24**	**0.50**	**0.52**	**0.00**	**0.38**	**0.74**	**0.63**

ARRIVE items: 1. Study design, 2. Sample size, 3. Inclusion and exclusion criteria, 4. Randomisation, 5. Blinding, 6. Outcome measures, 7. Statistical methods, 8. Experimental animals, 9. Experimental procedures, 10. Results, 11. Abstract, 12. Background, 13. Objectives, 14. Ethical statement, 15. Housing and husbandry, 16. Animal care and monitoring, 17. Interpretation/scientific implications, 18. Generalisability/translation, 19. Protocol registration, 20. Data access, 21. Declaration of interest.

**Table 2 pharmaceutics-16-01144-t002:** Drugs’ mechanisms of action.

Drug	Affinity					
	SERT	DA	5-HT	H	M	Others
**Fluoxetine**						
Prozac				H2		α2, α3, β4
Fluoxetine hydrochloride	SERT		5-HT2C			
Eli Lilly						
**Paroxetine**						
Seroxat			5-HT1			α1, α2, β
Paroxetine hydrochloride	SERT	D2	5-HT2	H1		
GlaxoSmithKline (GSK)			5-HT3			
**Sertraline**						
Zoloft						σ1
Sertraline hydrochloride	SERT					
Pfizer						
**Citalopram**						
Cipramil						
Citalopram hydrobromide	SERT					
Lundbeck				H1		
**Escitalopram**						
Lexapro			5-HT1A		M1	
Escitalopram oxalate	SERT	D2	5-HT2A	H1		
Lundbeck			5-HT2C			α2

## Data Availability

The original contributions presented in the study are included in the article/Appendix A, further inquiries can be directed to the corresponding authors.

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
