# Peer review of "Comparative Efficacy of Animal Depression Models and Antidepressant Treatment: A Systematic Review and Meta-Analysis"

_pharmaceutics, 2024, doi:10.3390/pharmaceutics16091144_

Round 1
Reviewer 1 Report
Comments and Suggestions for Authors
Dear authors,
Thank you for great work. It was pleasure to read yours manuscript. Methods were appropriate described and established. Introduction and results well organized and comprehensive. Limitation data presented.
I accept your manuscript in present form.
Author Response
Comment 1:
Thank you for great work. It was pleasure to read yours manuscript. Methods were appropriate described and established. Introduction and results well organized and comprehensive. Limitation data presented.
I accept your manuscript in present form.
Response 1:
We would like to express our gratitude to the Reviewer for his/her time and the dedication given in the positive review of our work. Thank you!
Reviewer 2 Report
Comments and Suggestions for Authors
After reading this manuscript I suggest for authors to decrease the tone of the discussion explaining the postitive effects of drugs in animal models based on a low number of publications. Althougth this is stated I do think that these section do not deserve this much focus.
In the discussion section you mention ‘….The highest efficacy was observed for paroxetine and sertraline; for the opposite side of the spectrum, there were citalopram and escitalopram…’ please elaborate this.
After changing this I think it is suitable for publciation.
Comments on the Quality of English LanguageNone
Author Response
We would like to express our gratitude to the Reviewer for the important comments, which have, of course, been taken into account to enhance the quality of our study. We will address each comment point by point. All changes in the document were made with the MS WORD 'track changes' option enabled.
Comment 1: After reading this manuscript I suggest for authors to decrease the tone of the discussion explaining the postitive effects of drugs in animal models based on a low number of publications. Althougth this is stated I do think that these section do not deserve this much focus.
Response 1: Following the Reviewer's suggestion, the discussion has been shortened. Attention has also been given to statements regarding the effectiveness of the studied drugs.
Comment 2: In the discussion section you mention ‘….The highest efficacy was observed for paroxetine and sertraline; for the opposite side of the spectrum, there were citalopram and escitalopram…’ please elaborate this.
Response 2: We would like to thank the Reviewer for highlighting this. Indeed, every drug in our forest plot analysis showed statistically significant results in their favour (as opposed to the control), so the use of the phrase 'opposite side of the spectrum' was unnecessarily misleading, suggesting that the results for citalopram and escitalopram did not confirm their effectiveness (which, of course, is not true). An appropriate correction was made:
“The meta-analysis showed that all tested drugs were effective in various ADMs described in our study. However, an essential factor that needs to be considered while assessing the results is that most of the included studies in our meta-analysis described the use, safety and efficacy of fluoxetine. At the same time, other medications were analysed based on a much smaller number of papers. That's why fluoxetine results could be the most precise even though other drugs showed more significant Mean Difference results (such as sertraline or paroxetine).”
Reviewer 3 Report
Comments and Suggestions for Authors
In this review article, Ratajczak and colleagues provide a comprehensive overview of the relationship between animal depression models and antidepressant treatment. They review 25 studies from several databases to analyze the popularity of both animal depression models and rat strains through meta-analysis. The authors report that the chronic unpredictable mild stress model and the Wistar and Sprague-Dawley rat strains are commonly used in preclinical depression research, affirming the efficacy of SSRIs, particularly fluoxetine, in reducing immobility time. Overall, the manuscript offers a thorough exploration of animal depression models and antidepressant research, delivering a well-written and detailed review for the field. However, several significant issues could improve the manuscript's quality:
-
The two major characteristics of depression-like behaviors are despair and anhedonia, which reflect the symptoms experienced by humans suffering from depression. However, the study is designed to review the efficacy of various animal depression models and selective serotonin reuptake inhibitors (SSRIs) based solely on immobility time. This single factor may not fully represent the core aspects of depression-like behavior. The reviewer suggests that the authors include additional parameters, such as anhedonia, to reassess the reports.
-
In the introduction, the authors provide background information about chronic unpredictable mild stress (CUMS). They describe CUMS as being restricted to rat models; however, this model also applies to mice. Therefore, the reviewer suggests that the authors replace references to rats with "rodents" in the introduction to better reflect the applicability of the model.
Numerous typos are present in the manuscript and should be rectified.
Author Response
We would like to express our gratitude to the Reviewer for the important comments, which have, of course, been taken into account to enhance the quality of our study. We will address each comment point by point. All changes in the document were made with the MS WORD 'track changes' option enabled.
Comment 1: The two major characteristics of depression-like behaviors are despair and anhedonia, which reflect the symptoms experienced by humans suffering from depression. However, the study is designed to review the efficacy of various animal depression models and selective serotonin reuptake inhibitors (SSRIs) based solely on immobility time. This single factor may not fully represent the core aspects of depression-like behavior. The reviewer suggests that the authors include additional parameters, such as anhedonia, to reassess the reports.
Response 1:
Thank you for highlighting this very important aspect, which we completely agree with. Clinical depression is a complex condition, and further simplifying the already simplified perspective on the animal model of this disease can be problematic. Relying solely on the immobility time parameter to confirm the effectiveness of the model and the antidepressant treatments is indeed an oversimplification. Introducing additional parameters, such as anhedonia, would certainly have enhanced the value of the study and broadened the perspective on this topic.
However, the decision to base our systematic review solely on immobility time and within very strict parameters of the test itself (considering there are various versions of the forced swimming test [Porsolt test]), was made at the beginning stage of our work. Information regarding this decision was published in December 2022 in the protocol of our study (https://osf.io/xvpq2).
Therefore, despite the Reviewer’s valid observation, considering the methodological complexity and the fundamental principles of conducting systematic reviews, we cannot expand the extracted results and, consequently, the meta-analysis to include anhedonia. This parameter was not considered during the literature search, screening of publications, or data extraction. To take this step would require us to restart the entire process, which, given the duration of this project, the number of studies analysed, and the multi-step nature of the process, is not feasible.
The decision to focus solely on immobility time was made primarily because it is the most commonly analysed and considered parameter in the context of animal models of depression. Additionally, using only one parameter significantly limited the number of publications we needed to analyse. We took a similar approach in the context of the drugs analysed, selecting only the five most widely discussed in the literature. Please also note that the phenomenon of anhedonia is frequently addressed in the discussion section of the manuscript.
However, as we have mentioned, this is an important aspect, and we have included a discussion of it in the 'Limitations' subsection of the manuscript, as the Reviewer’s comment pointed out a significant limitation of our study that we had not previously considered.
“Another important factor that significantly oversimplified the view on the animal model of depression is the exclusion of other objective parameters commonly used in the field of animal studies to confirm the effectiveness of depression models. One of the most important examples of this would be anhedonia, which was not analysed in our systematic review. By narrowing the inclusion criteria to only one parameter (immobility time), the perspective on the models and their response to treatment was simplified.”
Comment 2: In the introduction, the authors provide background information about chronic unpredictable mild stress (CUMS). They describe CUMS as being restricted to rat models; however, this model also applies to mice. Therefore, the reviewer suggests that the authors replace references to rats with "rodents" in the introduction to better reflect the applicability of the model.
Response 2: As the Reviewer suggested, we replaced the term “rats” with “rodents” or "animals" in the whole manuscript, if not only rats are used in a specific model.
Round 2
Reviewer 3 Report
Comments and Suggestions for Authors
The authors effectively addressed the reviewer's feedback and revised the manuscript accordingly, leading the reviewer to acknowledge their dedication and conclude that the manuscript is now suitable for acceptance in its current state.